# Factors Associated with Atopic Dermatitis and Allergic Rhinitis among Residents of Two Municipal Areas in South Korea

**DOI:** 10.3390/medicina55050131

**Published:** 2019-05-12

**Authors:** Dilaram Acharya, Bishnu Bahadur Bajgain, Seok-Ju Yoo

**Affiliations:** 1Department of Preventive Medicine, College of Medicine, Dongguk University, Gyeongju 38066, Korea; dilaramacharya123@gmail.com; 2Department of Community Medicine, Kathmandu University, Devdaha Medical College and Research Institute, Rupandehi 32907, Nepal; 3Department of Health Care Management, National Open College, Pokhara University, Lalitpur 44700, Nepal; abajgain99@gmail.com

**Keywords:** allergic disease, municipal area, housing type, South Korea

## Abstract

*Background and Objectives:* The growing burden and deleterious health consequences of allergic diseases, especially of allergic rhinitis (AR) and atopic dermatitis (AD), in developed countries remains an important public health issue. The current study aimed to assess the prevalence and to identify the risk factors of atopic dermatitis and allergic rhinitis among residents of Pohang-Si and Yeongdeok-Gun, two municipal areas in South Korea. *Materials and Methods*: A cross-sectional study was conducted in both municipal areas between 12 November and 13 December 2017. A total of 302 subjects were recruited from 100 households (25 apartments and 25 houses in each municipality), by system extraction according to district code numbers. Data were collected using International Study of Asthma and Allergies in Childhood (ISAAC) Standard Questionnaires for children and a health questionnaire for adults. Risk factors were identified by multivariate logistic regression analysis. *Results*: Of the 302 study participants, 12.9% and 25.5% had AD and AR, respectively. The significant factors associated with AD by multivariate logistic regression analysis were age ≥19 years (aOR (adjusted odds ratio) 6.9; 95% CI (confidence interval) (2.9–16.37)), residence in Pohang-Si (aOR 2.5; 95% CI (1.18–5.53)), and family history of allergic disease (aOR 2.3; 95% CI (1.09–4.9)). Similarly, the significant factors associated with AR were male gender (aOR 2.3; 95% CI (1.24–4.42)), age ≥19 years (aOR 4.4; 95% CI (2.28–8.48)), residence in Pohang-Si (aOR 2.8; 95% CI (1.51–5.37)), and family history of allergic disease (aOR 6.7; 95% CI (3.50–12.82)). *Conclusion:* The present study shows that age ≥19 years, residence in Pohang-Si, and family history of allergic disease are risk factors for AD and AR, and that, additionally, male gender is a risk factor of AR. Understanding the risk factors of allergic diseases can aid the design and implementation of evidence-specific strategies to reduce the long-standing problems associated with allergic disease.

## 1. Introduction

Allergic diseases represent growing health and economic burdens worldwide [1,2], and have frequently been reported to impair quality of life and retard cognitive functions [3,4]. About 40% of the global population suffer from an allergic disorder, and of the many allergic diseases known, the most common are atopic dermatitis (AD), allergic rhinitis (AR), and asthma [3,5]. The common clinical features of AR include profuse watery rhinorrhea, sneezing, itchy nose, and congestion with occasional experiences of itchy conjunctiva, ears, and throat [6]. On the other hand, AD is a chronic relapsing inflammatory skin disease accompanied by respiratory allergy, recurrent bacterial (impetigo), fungal (tinea), and viral (Herpes Simplex molluscum contagiosum) skin infections [7,8,9,10]. Interestingly, a recent systematic review indicated that patients with AD have increased risks of cardiovascular diseases, certain malignancies, autoimmune diseases, and neuropsychiatric diseases [11]. Furthermore, the coexistence of asthma, allergic rhinitis, and eczema gives rise to more severe and intense symptoms which may lead to a poor quality of life [12]. 

Air pollution, climate change, and global warming are factors that underlie the increasing prevalence of allergic diseases, and the higher level of air pollution in developed countries has also been reported to be positively associated with allergic diseases [13,14]. According to the 5th Korea National Health and Nutrition Examination Survey, the prevalence of atopy and AR was 31.2 and 17.2%, respectively [15]. Another large-scale South Korean study performed in children in 2012 revealed a 12-month prevalence of allergic rhinitis of 23.9%, 30%, 31.4%, and 34.2% among pre-school, elementary school, middle school, and high school students, respectively, and a prevalence of "atopic dermatitis during the last 12 months" of 19%, 17.4%, 12.3%, and 11.4%, respectively, among the same children [16]. In a recent population-based Korean study, at a younger age, asthma, atopic dermatitis, and a family history of allergic disease were identified as important risk factors of AR [17], whereas in another recent population-based Korean study, sex was found to be a risk factor of atopy and sex, age, marital status, and stress level were identified as risk factors of AR [15]. Despite the existence of some previous nationwide studies regarding allergic diseases in South Korea, local and small area studies are equally important to gaining an insight, filling in gaps in literature and planning and conducting large area studies. In addition, studies presenting the results of prevalence and risk factors of allergic diseases for both children and adults, and for rural and urban settings in the South Korean context have rarely been performed. In view of the increasing industrialization in South Korea and the increasing prevalence of allergic disease, the current study was undertaken to assess the prevalence and identify the risk factors of atopic dermatitis and allergic rhinitis among residents of Pohang-Si and Yeongdeok-Gun, which are two municipal areas in South Korea.

## 2. Materials and Methods

### 2.1. Study Design, Sampling, and Setting

This cross-sectional study was conducted in Pohang-Si and Yeongdeok-Gun between 12 November and 13 December 2017. To adjust for the effect sizes of residential area and housing type on allergic disease, we selected Pohang-Si (population ≥500,000) as a representative urban area, and Yeongdeok-Gun (population <50,000) as a representative rural area. To address housing type, one apartment area and one housing area were arbitrarily selected in each area. Fifty households (25 apartments and 25 houses) were selected in each area by system extraction using district code numbers. Surveyed households were individually visited and information on the residents was obtained using questionnaires. We aimed to enroll at least one participant below 19 years of age and one above 19 years of age from the selected households. All eligible participants were recruited when more than two participants were found in a household. Study participants who had experienced respiratory infection within the months preceding the study or had a history of smoking were excluded from the study.

### 2.2. Data Collection

For data collection, we adapted a questionnaire from the Korean version of the International Study of Asthma and Allergies in Childhood (ISAAC) Standard Questionnaire for children, and we designed a self-developed questionnaire for adults in consultation with experts such as medical doctors specializing in allergies and clinical immunology and an epidemiologist. We also adopted information for a questionnaire for adults based on a previously published paper from a South Korean study that made use of questions similar to the ISAAC Standard Questionnaires [18]. Survey items included general items such as age, gender, and type of residence, and details of allergic diseases such as allergic rhinitis, atopic dermatitis, and details of other factors likely to be associated with allergic diseases [19]. Questionnaires were administered by pertained interviewers, and interviews were conducted after obtaining informed consent.

### 2.3. Definitions of Variables

The main outcome variables were the presence of atopic dermatitis and allergic rhinitis, which was determined by the responses to standardized parent- and self-reported questions, and the doctor’s diagnosed evidence. Furthermore, a study participant was considered to have had AR as defined by having a runny, itchy or stuffed nose without a cold in the last 12 months, and AD as defined by having an itchy rash that persisted for at least 6 months and was located in the antecubital or popliteal fossae, wrists, ankles, neck or face during the last 12 months [20]. Gender, age, area of residence, housing type, and family history of allergic disease were defined as independent variables. Gender was categorized as male or female, age as <19 years or ≥19 years, area of residences as Pohang-Si or Yeongdeok-Gun, housing type as apartment or house, and family history of an allergic disease (both parents) as yes or no.

### 2.4. Ethical Considerations

We assert that ethical issues such as data fabrication, data falsification, and double publication were properly addressed. Since this study falls within the rules and regulations of the Korean Bioethics and Safety Act (Article, 15 (2)) set by the Ministry of Health and Welfare of South Korea, additional approval from an institutional review board or ethical committee is not necessary. All study subjects provided informed consent before study recruitment, and all personal identifiers were removed prior to data analysis.

### 2.5. Statistical Analysis

Questionnaire responses were coded and entered into Excel. Statistical analysis was performed using SPSS for Windows (version 22.0, SPSS Inc. Chicago, IL, USA). Univariate analysis was performed using the chi-square test to subject general characteristics and environmental factors as independent variables. Categorical variables with significant differences in the experience of allergic disease diagnoses (*p* value < 0.1) were entered into the multivariate analysis. Odds ratios (ORs) are reported with 95% confidence intervals (CIs), and statistical significance was accepted for *p* values < 0.05. 

## 3. Results

### 3.1. Status of Common Allergic Diseases and Characteristics of Subjects 

Table 1 summarizes the status of the two selected allergic diseases and the participants’ personal characteristics. Of the 302 study subjects, 12.9% and 25.5% had atopic dermatitis or allergic rhinitis, respectively. Of the 132 subjects from Pohang-Si, 20.5% and 38.6% had atopic dermatitis or allergic rhinitis, respectively, whereas of the 170 from Yeongdeok-Gun, only 7.1% and 15.3% had atopic dermatitis or allergic rhinitis. About half (49.3%) of the study subjects were males. The majority (60.9%) of the study subjects were >19 years old, 56.3% were from Yeongdeok-Gun, 53% lived in an apartment, and 60.3% had no family history of allergic diseases. Univariate analysis showed that age, area of residence, and family history of allergic disease were significantly associated with atopic dermatitis, and that gender, age, area of residence, and family history of allergic disease were significantly associated with allergic rhinitis (Table 1).

### 3.2. Factors Associated with Atopic Dermatitis and Allergic Rhinitis 

Table 2 presents the results of the multivariate analyses of factors associated with atopic dermatitis or allergic rhinitis among residents of the two municipal areas. Age ≥19 years, residing in Pohang-Si, and a family history of allergic disease resulted in higher odds ratios of having atopic dermatitis. Similarly, male gender, age ≥19 years, residing in Pohang-Si, and a family history of allergic disease resulted in higher odds ratios of allergic rhinitis.

Study subjects aged ≥19 years were 6.9 times more likely (aOR 6.9; 95% CI (2.9–16.37)) to have atopic dermatitis than those aged <19 years. Those residing in Pohang-Si were 2.5 times more likely (aOR 2.5; 95% CI (1.18–5.53)) to have atopic dermatitis than those residing in Yeongdeok-Gun. Likewise, study participants with a family history of allergic disease were 2.3 times more likely (aOR 2.3; 95% CI (1.09–4.9)) to have atopic dermatitis. On the other hand, males were 2.3 times more likely (aOR 2.3; 95% CI (1.24–4.42)) to have allergic rhinitis than females, those aged ≥19 years were 4.4 times more likely (aOR 4.4; 95% CI (2.28–8.48)) to have allergic rhinitis than those aged <19 years, those living in Pohang-Si were 2.8 times more likely (aOR 2.8; 95% CI (1.51–5.37)) to have allergic rhinitis than those living in Yeongdeok-Gun, and those with a family history of allergic disease were 6.7 times more likely (aOR 6.7; 95% CI (3.5–12.82)) to have allergic rhinitis than their counterparts.

## 4. Discussion

This study revealed that of the 302 study subjects, 12.9 % and 25.5% had AD and AR, respectively. Of the 132 subjects from Pohang-Si, 20.5% and 38.6% had AD and AR, respectively, while only 7.1% and 15.3% of the 170 subjects from Yeongdeok-Gun had AD and AR. In addition, subjects aged ≥19 years, those residing in Pohang-Si, and those with a family history of allergic disease had higher odds ratios of AD and AR, while male gender had a higher odds ratio of AR.

Two previous nationwide studies conducted in South Korea reported AR prevalence of 27% and 17.2 % [15,17], which concur with our observations. In one of these previous nationwide studies, 31.2% of the study population suffered from AD [15], which is more than twice that encountered in the present study. This may have been because Pohang-Si is an industrial city and industrial pollutants might have influenced such rising prevalence. However, the prevalence of AR and AD found in the present study is in line with that observed in a nationwide Korean study. Our multivariate analysis also revealed that Pohang-Si had higher odds ratios of AD and AR than Yeongdeok-Gun. In fact, several studies conducted in developed countries have shown a higher prevalence of allergic diseases [12,21,22,23,24], and in another, urban areas were found to be more prone to allergic diseases [25], which supports the argument that industrialization is associated with the higher prevalence of allergic diseases.

Views differ regarding whether age is a risk factor of allergic diseases [26,27]. One study [21] demonstrated a weak relation between age and allergic diseases, whereas another study [22] found a positive association. In the present study, those aged ≥19 had higher odds for AD and AR, possibly because our subjects were more exposed to allergens because this population may be more ambulatory, socio-economically active and might encounter more chances of being exposed to allergens during work and travel compared to the lower age group population.

A number of studies have demonstrated an association between family history of allergic disease and the occurrence of atopic diseases in offspring [12,17,28,29,30]. Genetic contributions to allergic disease were estimated to be greater than 50% in two studies, with a range of 36–79% genetically inherited [31,32]. Genetic susceptibility for allergic diseases has been well reported, suggesting the common and distinct genetic loci associated with these diseases [33]. We found that male gender was significantly associated with AR, which is consistent with the results of other studies [24,25], but contrary to the findings of Gough et al. [12]. However, in addition to exposure to the environmental allergens, sex-specific genetic effects might also have contributed to the gender differences in the occurrence of allergic diseases because there are underlying differences in the inflammatory pathways to different allergens between women and men [34,35,36,37]. 

Despite our efforts, the study has some limitations that require consideration. Firstly, we reported the prevalence and associated risk factors of only two allergic diseases as determined by subject self-reporting and we lacked information about non-allergic diseases. Furthermore, the method used did not rule out the co-existence of AR, AD, and other allergic diseases, and did not provide information on the types of allergens. Secondly, a relatively small number of subjects were enrolled, and thus, the generalizability of our findings should be considered with caution. Nevertheless, we believe that our findings add to current knowledge and aid the design of customized preventive strategies, such as the creation of general awareness about allergic diseases with an intensive focus on risk groups.

## 5. Conclusions

This study showed that age of ≥19 years, residence in Pohang-Si, family history of allergic disease, and male gender were found to be factors significantly associated with AD and AR, and in addition, male gender was found to be significantly associated with AR. Understanding the risk factors of allergic diseases aids the design and implementation of evidence-based strategies aimed at reducing the long-standing problem of allergic disease. Further studies are required to identify the common urban allergens and other risk factors responsible for atopic dermatitis and allergic rhinitis.

## Figures and Tables

**Table 1 medicina-55-00131-t001:** Prevalence of atopic dermatitis and allergic rhinitis with respect to the personal characteristics of participants from Pohang-Si and Yeongdeok-Gun.

Variables	Total *n* (%)	Common Allergic Diseases (Yes, %)	
	302 (100)	Atopic Dermatitis Yes, *n* (%) 39 (12.9)	Allergic Rhinitis Yes, *n* (%) 77 (25.5)
**Gender**		*p* = 0.266 *	*p* = 0.008 *
Male	149 (49.3)	16 (10.7)	48 (32.2)
Female	153 (50.7)	23 (15)	29 (19)
**Age group (in years)**		*p* ≤ 0.001 *	*p* ≤ 0.001 *
<19 years	184 (60.9)	8 (4.3)	28 (15.2)
≥19 years	118 (39.1)	31 (26.3)	49 (41.5)
**Area of residence**		*p* = 0.001 *	*p* ≤ 0.001 *
Pohang-Si	132 (43.7)	27 (20.5)	51 (38.6)
Yeongdeok-Gun	170 (56.3)	12 (7.1)	26 (15.3)
**Housing types**		*p* = 0.360 *	*p* = 0.205 *
Apartment	160 (53)	18 (11.3)	36 (22.5)
Housing	142 (47)	21 (14.8)	41 (28.9)
**Family history of having allergic diseases**		*p* = 0.003 *	*p* ≤ 0.001 *
No	182 (60.3)	15 (8.2)	20 (11)
Yes	120 (39.7)	24 (20)	57 (47.5)

* Chi-square test used; *p* values of <0.05 were considered statistically significant.

**Table 2 medicina-55-00131-t002:** Adjusted odds ratio as determined by multivariate logistic regression analysis for the prevalence of atopic dermatitis and allergic rhinitis with respect to the personal characteristics of the residents of Pohang-Si and Yeongdeok-Gun.

Variables	Risk Factors for Common Allergic Diseases (aOR, 95% CI)	
	Atopic Dermatitis	Allergic Rhinitis
**Gender**		*p* = 0.009
Female		1 (Ref.)
Male		2.3 (1.24–4.42)
**Age group (in years)**	*p* ≤ 0.001	*p* ≤ 0.001
<19 years	1 (Ref.)	1 (Ref.)
≥19 years	6.9 (2.9–16.37)	4.4 (2.28–8.48)
**Area of residence**	*p* = 0.018	*p* = 0.001
Yeongdeok-Gun	1 (Ref.)	1 (Ref.)
Pohang-Si	2.5 (1.18–5.53)	2.8 (1.51–5.37)
**Family history of having allergic diseases**	*p* = 0.028	*p* ≤ 0.001
No	1 (Ref.)	1 (Ref.)
Yes	2.3 (1.09–4.9)	6.7 (3.50–12.82)

aOR = adjusted odds ratio, Ref = reference, CI = confidence interval; *p* values of <0.05 were considered statistically significant; odds ratios are adjusted by sex, age group, area of residence, and family history of allergic diseases.

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
