# Peer review of "Factors Associated with Atopic Dermatitis and Allergic Rhinitis among Residents of Two Municipal Areas in South Korea"

_medicina, 2019, doi:10.3390/medicina55050131_

Round 1

Reviewer 1 Report

Journal: Medicina

Manuscript ID: medicina-460298

Title: Factors Associated with Atopic Dermatitis and Allergic Rhinitis Among Residents of Two Municipal Areas in South Korea

General comments

Thank you for asking me to review this article aimed to assess the prevalences and to identify the risk factors of atopic dermatitis and allergic rhinitis among residents of Pohang-Si and Yeongdeok-Gun. This is definitively an interesting field of investigation and worthwhile to study. However, I have some concerns over the presentation of the work. Please, see the comments for authors.

Major comments

Introduction:

The introduction is well written, but it does not inform the reader on the rationales which have led the authors to conduct this study. Probably, authors should rewrite this part so that the “introduction section” can better argue/support the aim of the study as well as the need to get information on this issue. Please, improve this section also referring to: Ann Allergy Asthma Immunol. 2017 Nov;119(5):446-451; Allergy Asthma Proc. 2007 Jan-Feb;28(1):40-3

I have several doubts about the methodological aspects:

Study design:

It would be better to specify the time interval that was analyzed (from…to…)

Authors have considered as eligible studies only published studies. Why did they not also include unpublished data? The choice made by the authors could significantly affect the results, leading to significant data loss.

Eligibility criteria:

Study should provide more details on the “types of participants”:

What were the demographic and clinical characteristics of the enrolled patients?

According to which criteria have been diagnosed atopic dermatitis and allergic rhinitis? Please, specify them.

Data analysis:

Was sample size calculation performed?

The discussion should be more focused on the results of this study, comparing them to the current literature knowledge.

Minor comments

The entire manuscript would require to be edited by an expert in English grammar to correct grammar and typos.

Author Response

First Reviewer-First Round

Journal: Medicina

Manuscript ID: medicina-460298

Title: Factors Associated with Atopic Dermatitis and Allergic Rhinitis Among Residents of Two Municipal Areas in South Korea

General comments

Thank you for asking me to review this article aimed to assess the prevalences and to identify the risk factors of atopic dermatitis and allergic rhinitis among residents of Pohang-Si and Yeongdeok-Gun. This is definitively an interesting field of investigation and worthwhile to study. However, I have some concerns over the presentation of the work. Please, see the comments for authors.

Response: We thank you very much and highly appreciate reviewer’s valuable comments and suggestions. We have revised the manuscript based on reviewers’ comments and suggestions. All changes are marked with blue colored writing in this revised version of the manuscript to allow reviewers’ verifications.

Major comments

Comment: Introduction: The introduction is well written, but it does not inform the reader on the rationales which have led the authors to conduct this study. Probably, authors should rewrite this part so that the “introduction section” can better argue/support the aim of the study as well as the need to get information on this issue. Please, improve this section also referring to: Ann Allergy Asthma Immunol. 2017 Nov;119(5):446-451; Allergy Asthma Proc. 2007 Jan-Feb;28(1):40-3

Response: Agree. We appreciate reviewer’s comments and useful published papers suggested. We have revised the introduction as suggested.

I have several doubts about the methodological aspects:

Study design:

Comment: It would be better to specify the time interval that was analyzed (from…to…)

Response: Agree. Corrected.

Comment: Authors have considered as eligible studies only published studies. Why did they not also include unpublished data? The choice made by the authors could significantly affect the results, leading to significant data loss.

Response: Agree. For your kind information, we do have almost all data set online as disease portal in South Korea. We have tried to utilize all of them as for as we have availed them.

Eligibility criteria:

Comment: Study should provide more details on the “types of participants”:

Response: Agree. We recruited all study participants with the aim that at least one above 19 years and one below 19 years of age groups which has now been mentioned in the current version of the manuscript.

Comment: What were the demographic and clinical characteristics of the enrolled patients? According to which criteria have been diagnosed atopic dermatitis and allergic rhinitis? Please, specify them.

Response: Agree. We have revised the method section with more clear information on how outcome variables and independent variable s were defined.

Data analysis:

Comment: Was sample size calculation performed?

Response: Agree. The number of samples was calculated using G Power 3.1 program*. The 88 samples were required when the effect size was set to 0.3, α = 0.05, and the power was 80% based on the X2 test. And 138 samples were required when the effect size was 0.15, α = 0.05, the power was 80% and the number of predictors = 5; based on the linear multiple regression analysis. So, the number of samples we have obtained in this study is sufficient to perform the necessary statistical analysis.

*Ref) Faul, F., Erdfelder, E., Buchner, A., & Lang, A.-G. (2009). Statistical power analyses using G Power 3.1: Tests for correlation and regression analyses. Behavior Research Methods, 41, 1149-1160.

Comment: The discussion should be more focused on the results of this study, comparing them to the current literature knowledge.

Response: Agree. We have revised the discussion section of the manuscript as suggested.

Minor comments

Comment: The entire manuscript would require to be edited by an expert in English grammar to correct grammar and typos.

Response: Agree. We have thoroughly checked and edited in a number of spaces and also we sought opinion of native English speaker after we completed our revision of the manuscript.

Reviewer 2 Report

This is a well written study. The findings of the authors are modest and not novel, similar findings have been obtained elsewhere in Korea. However, the rural/urban difference is of interest.

The authors have used the ISAAC questionnaire for children, I presume anyone younger than 19 years while they developed their own questionnaire for adults. Where the adult questions for symptoms the same as the ISAAC questions? I would like to have seen the adult questions for rhinitis and eczema symptoms. Was the adult questionnaire validated?

I presume family history was for both parents? This needs to be stated.

Why is there no data for asthma prevalence between the urban/rural groups. The ISAAC questionnaire certainly could easily have given this data and would have strengthened this paper.

I question why no local ethical approval was not required, I was unaware that the Korean Academy of Asthma Allergy and Clinical Immunology is able to grant ethical approval. The authors need to clarify this.

Author Response

Second Reviewer-First Round

Comments and Suggestions for Authors

This is a well written study. The findings of the authors are modest and not novel, similar findings have been obtained elsewhere in Korea. However, the rural/urban difference is of interest.

Response: Thank you very much for the encouragement. We have revised the manuscript extensively based on reviewers’ comments and suggestions. All changes are marked with blue colored writing in this revised version of the manuscript to allow reviewers’ verifications.

Comment: The authors have used the ISAAC questionnaire for children, I presume anyone younger than 19 years while they developed their own questionnaire for adults. Where the adult questions for symptoms the same as the ISAAC questions? I would like to have seen the adult questions for rhinitis and eczema symptoms. Was the adult questionnaire validated?

Response: Agree. Yes, the adult questions were almost all same as the ISAAC questions.  We have now cited the previous Korean study that made use of both for children and adult questionnaires, for your kind information.

Comment: I presume family history was for both parents? This needs to be stated.

Response: Agree. We have now mentioned about it in the manuscript.

Comment: Why is there no data for asthma prevalence between the urban/rural groups. The ISAAC questionnaire certainly could easily have given this data and would have strengthened this paper.

Response: Strongly agree. We had the data on Asthma too. Unfortunately, the prevalence of asthma was minimal and none of result was significant to be reported. Therefore, we decided to only present the result of AD and AR.

Comment: I question why no local ethical approval was not required, I was unaware that the Korean Academy of Asthma Allergy and Clinical Immunology is able to grant ethical approval. The authors need to clarify this.

Response: Agree. We have had prior communication about it to the journal authority and reached to consensus that we must mention in written statement about how ethics can be exempted for this study. To be honest, since this study falls within the rules and regulations of Korean Bioethics and Safety    Act  (Article, 15 (2)) set by Ministry of Health and Welfare of South Korea, therefore, an additional  approval from Institutional review board or ethical committee is not necessary. We have now mentioned the same in the current version of the manuscript replacing the previous statement.

Reviewer 3 Report

Dear authors, thank you for this intersering paper. 

My overall comment is that the discussion should be enriched by other hypothesis that may explain the findings. 

Please find below some specific comments: 

Abstract:

-          line 21: rephrase (example: using ISAAC standard questionnaires for children and a health questionnaire for adults)

-          line 23 and line 109: Doses it mean that 12.9% had AD only (no AR) and 25.5% had AR only (no AD)?

-          line 27: please add “with” as follow: associated with AR

Introduction

-          line 48: please correct the sentence (makes is associated?)

-          line 51: please correct the sentence (have also be?)

-          line 58: “at” a younger age

-          line 60: please add a coma between “study” and “sex”

Material and methods

-          line 88: why has age been categorized as <19 and >19 ? I would suggest a more categorized variable in order to highlight the difference between children, adolescent, adults and elderly people for example.

Discussion

-          lines 162-163: I do not understand the interpretation of the results regarding age. Why subjects aged more than 19 y may be more exposed to allergens?

-          Line 167: please correct the sentence “we found that … “

-          Line 169: “we suggest that”

-          Line 169: what about genetics?

-          Line 176: “aid the design of customized preventive strategies”: such as? Almost all risk factors that were found are not modifiable fators (age, gender, family history), so we cannot apply preventive measures

Conclusion

-          Line 178: no need to repeat the prevalence results

Author Response

Third Reviewer-First Round

Dear authors, thank you for this interesting paper.  My overall comment is that the discussion should be enriched by other hypothesis that may explain the findings. Please find below some specific comments: 

Response: We thank you very much. We highly appreciate the reviewer’s comments and suggestions. We have attempted to extensively modify the manuscript based on reviewers’ comments and suggestions in this revised version of the manuscript. For your kind information, we have marked all revisions made with blue colored writing to allow reviewers’ verification.

Abstract:

Comment:  line 21: rephrase (example: using ISAAC standard questionnaires for children and a health questionnaire for adults)

Response: Agree. Corrected.

Comment: line 23 and line 109: Doses it mean that 12.9% had AD only (no AR) and 25.5% had AR only (no AD)?

Response: Agree. Corrected.

Comment: line 27: please add “with” as follow: associated with AR

Response: Agree. Corrected.

Introduction

Comment: line 48: please correct the sentence (makes is associated?)

Response: Agree. Sentence corrected.

Comment: line 51: please correct the sentence (have also be?)

Response: Agree. Sentence corrected.

Comment: line 58: “at” a younger age

Response: Agree.  Corrected.

Comment: line 60: please add a coma between “study” and “sex”

Response: Agree. Comma added as suggested.

Material and methods

Comment: line 88: why has age been categorized as <19 and >19? I would suggest a more categorized variable in order to highlight the difference between children, adolescent, adults and elderly people for example.

Response:  Agree. Honestly, we collected information to identify the overall prevalence and risk factors between the study participants who below and above 19 years of age so that information on outcome can be gathered by two sets of questionnaires ( below 19 years-using ISSAC questionnaire and above 19 years self-developed questionnaires based on previously published paper)

Discussion

Comment:   lines 162-163: I do not understand the interpretation of the results regarding age. Why subjects aged more than 19 y may be more exposed to allergens?

Response: Agree. Comment has been addressed.

Comment: Line 167: please correct the sentence “we found that … “

Response: Agree. Sentence corrected.

Comment: Line 169: “we suggest that”

Response: Agree. corrected.

Comment: Line 169: what about genetics?

Response: Thank you very much for this comment. We have now added the logic with reference.

Comment: Line 176: “aid the design of customized preventive strategies”: such as? Almost all risk factors that were found are not modifiable fators (age, gender, family history), so we cannot apply preventive measures.

Response: Agree. Thank you very much for a very useful input.  We focused on creation of general awareness of allergic diseases especially to risk groups in this regard in the current version of the manuscript.

Conclusion

Comment: Line 178: no need to repeat the prevalence results

Response: Agree. Get corrected as suggested.

Round 2

Reviewer 2 Report

The authors have addressed my comments to my satisfaction.

Reviewer 3 Report

Dear authors,

Thank you for having taken my comments into account.